



# The development of a reference corner cube inertial suspension device

Bing Zhang[1], Xiaoyi Zhu[1,*], Qiong Wu[2,*], Bing Xue[1], Lili Xing[3], Yanxiong Wu[3],
Peng Su[1], Xiaolei Wang[1],Yuru Wang[1], Shuaibo Zhao[1]

[1] Institute of Earthquake Forecasting, China Earthquake Administration, Beijing, China.
[2] Institute of Geophysics, China Earthquake Administration, Beijing, China.
[3] Institute of Disaster Prevention, Sanhe Hebei, China.

*Correspondence to*: Xiaoyi Zhu (zxy_bj2008@126.com) ; Qiong Wu (wuqiong@cea-igp.ac.cn)

**Abstract**: The seismometer synchronous observation and zero crossing methods are applied to laser interferometer absolute gravimeter to suppress the vibration interference. However, during the synchronous observation of the seismometer and the gravimeter, the observation point of the seismometer does not coincide with the reference corner cube in space, resulting in spatial dislocation,

which cannot accurately reflect the vibration state of the reference corner cube. So, it is necessary to hang the reference corner cube on the elastic element to directly measure its vibration acceleration measurement. In this paper, an open-loop reference corner cube inertial suspension device(RCCISD) hanging the reference corner cube was developed based on the principle of seismometer, which is used to measure the vibration acceleration of the reference corner cube of the laser interferometer absolute

gravimeter.     Experimental test results show that the power spectrum of gravitational acceleration calculated by an interference fringe observed jointly by the RCCISD is about 40dB lower than that of the reference corner cube directly placed on the ground. RCCISD can restrain the vibration interference to a certain extent, not only can it measure the reference corner cube vibration more accurately than the seismograph synchronous observation method for the vibration compensation of gravity measurement,

but also the volume is about 1 / 3 of the Super-Spring volume, which can greatly reduce the height of the gravimeter.

**Keywords**: absolute gravimeter; Laser interferometer; vibration interference; the reference corner cube; inertial suspension device

## 1 Introduction

During the falling process of the free-falling corner cube in absolute gravimeter, the change of the displacement between the free-falling corner cube and the reference corner cube forms a fringe signal. The falling trajectory of the free-falling corner cube relative to the reference corner cube can be reconstructed by extracting the zero-crossing information of the fringe signal. The gravity acceleration of the measuring point can then be obtained using the least-squares fitting of the trajectory(T.

M.Niebauer, et al. 1995; Wu Shu-qing, 2012; HU H, et al. 2012). It is necessary to keep the reference corner cube stationary or its motion linear during the falling process to ensure the accuracy of the obtained gravitational acceleration. However, due to the environmental vibration and the free vibration



of the free-falling corner cube control system, the reference corner cube presents a complex vibration mode during the free fall of the free-falling corner cube, causing errors to the measurement results(Wu

Qiong, et al. 2012; Long Jianfeng, 2012).

In 1993, USGS published a report on observation and Model of earthquake background noise (Peterson,J.R.,1993). The report gives the results of power spectrum analysis of normal earth background noise from many seismic stations around the world, and gives a new model of high earth noise (NHNM) and a new model of low earth noise (NLNM). The peak noise of the acceleration power

spectrum in the $0.04Hz$~$1Hz$ band of the model comes from the interference of ocean waves, while the high-frequency noise above 1Hz mainly comes from human activities, wind and other factors. Because the ocean area accounts for about 70% of the earth's surface, the wave interference frequency is low and the propagation attenuation is slow, wave interference has become an unavoidable interference factor in absolute gravity measurement. In economically developed areas, human activity interference

may also reach a very high range, which has become the primary factor affecting the accuracy of gravity measurement.

Most of the free fall stroke in the absolute gravimeter is about 20cm, and the free fall time is about 0.2s. It takes several seconds to carry out a measurement, and the repeated measurement period is generally about 10s (T.M.Niebauer et al., 2011; Wu Qiong et al., 2017). For example, the FG5 absolute

gravimeter can make 200 measurements in 30 minutes with an average interval of 9s (T. M.Niebauer, et al. 1995). If the repeated measurement period is 10s, that is, the sampling rate is 0.1Hz, according to the sampling theorem, any signal and interference whose frequency is higher than 0.05Hz will be superimposed into the measured data in the form of frequency aliasing and become the largest source of random interference. The measurement accuracy is 2 $\mu Gal$ / $Hz$, corresponding to the longitudinal

coordinate-154dB in figure 1, it can be seen that the amplitude of wave interference may be 1000 times higher than the required measurement accuracy. Therefore, it is an important task in absolute gravity measurement to suppress the interference noise such as wave interference, man-made vibration interference and wind disturbance whose frequency is higher than that of $0.05Hz$.

At present, the FG5 series absolute gravimeter of Micro-GLaCoste Company of the United States is the

only commercial high-precision absolute gravimeter with a measurement accuracy of $2\mu Gal/\sqrt{Hz}$($2 \times 10^{-8}$m/s$^2$)and an accuracy of $2\mu Gal/\sqrt{Hz}$ (quiet site) (Micro-gLaCoste,2015). The FG5 absolute gravimeter adopts SuperSpring vibration isolation system and has a two-stage spring structure. The basic principle is that the position change of the reference corner cube relative to the auxiliary spring vibration isolation frame is detected by laser, and the upper end of the main spring is adjusted by active

feedback compensation to make the secondary spring vibration isolation frame follow the motion of the reference corner cube. After feedback compensation, it is equivalent to an ultra-long period spring damping suspension system with a period of 30 seconds, so as to attenuate low frequency vibration interference and restrain the influence of ground pulsation on absolute gravity measurement (YAO Jia-min, et al., 2019).



The free-falling corner cube driving mechanism of the Age-110 absolute gravimeter developed by the Institute of Geophysics, China Earthquake Administration, is mainly composed of gears, rack, carriage, guide rail, guide pillar, vacuum sealing parts, auxiliary support parts, servo motor, and the control system. The carriage is the free-fall carrier. The servo motor controls the carriage's lifting and release through the bite of the gear and the rack. The guide rail and rail strut are utilized to keep the carriage

moving up and down along the vertical direction. The lifting height of the carriage is about 18 cm. The free-falling time of the free fall is about 0.2 s. The motion process of the carriage mainly includes uniform lifting, pausing for 10 seconds, accelerating fall, uniform falling, decelerating falling, and resting(Wu Qiong, 2011; Wu Qiong, et al. 2012; Li Zhe, 2016).

  With the acceleration of the falling, the bite between the motor gear and the rack produces the moment

of inertia under the tension action, leading to high-speed motion friction, while the carriage and the guide rail also produce high-speed motion friction. In deceleration of the falling, the motor produces a reverse moment of inertia and motion friction. When the carriage displacement is equivalent to that of the free-falling corner cube, the two begin to contact to form an elastic collision, resulting in a specific inertia impulse. The gravimeter transmits the vibration caused by high-speed friction and elastic

collision to the ground, causing the vibration of the reference corner cube by the ground coupling. The vibration interference runs through the whole process of free flight of the free-falling corner cube, superimposing on the whole laser interference fringes to affect the accuracy of gravity measurement. Therefore, the seismometer synchronous observation and zero crossing methods are applied to Age-110 to suppress the vibration interference. Age-110 uses the seismometer to place the seismometer at the

5cm next to the reference corner cube for synchronous vibration observation and zero-crossing dotting algorithm for vibration compensation to achieve the purpose of vibration suppression.

  However, during the synchronous observation of the seismometer and the gravimeter, the observation point of the seismometer does not coincide with the reference corner cube in space, resulting in spatial dislocation, which cannot accurately reflect the vibration state of the reference corner cube. So, it is

necessary to hang the reference corner cube on the elastic element to directly measure its vibration acceleration measurement for real-time compensation.

  In this paper, an open-loop reference corner cube inertial suspension device (RCCISD) hanging the reference corner cube was developed based on the principle of seismometer, which is used to measure the vibration acceleration of the reference corner cube of the laser interferometer absolute gravimeter.

Afterwards, the reference corner cube inertial suspension device was deployed to the Age-110 to test the function and the performance. RCCISD can not only measure reference corner cube vibration more accurately than seismograph synchronous observation method for vibration compensation of gravity measurement, but also is smaller than super-long spring and can greatly reduce the height of gravimeter.

**2 Principle model**

In order to directly measure and analyze the vibration data of the reference corner cube, the reference corner cube can be elastically suspended on the pendulum of the open-loop seismometer to design a new open-loop reference corner cube inertial suspension device. The motion of the pendulum is measured by an electromagnetic transducer. A coil is installed on the pendulum and embedded in the

gap magnetic field of the magnetic steel. The induced electromotive force is generated by the movement of the coil in the magnetic field, and the output of the voltage signal is used to reflect the acceleration of the frame to the ground motion. The principal model is shown as figure.

According to Faraday's law, the induced electromotive force produced by the coil in the loop is $e_v = d\Phi / dt$, which $d\Phi$ is the variation of magnetic flux. When the coil has N turns, the radius of the coil is r

and the magnetic induction intensity is B. when the center of the coil moves away from dx:

$$d\Phi = 2\pi r N B dx$$    (1)

Therefore, the induced electromotive force can be written as

$$e_v = 2\pi r N B \frac{dx}{dt} = U \frac{dx}{dt}$$    (2)

Where U is the voltage sensitivity, the larger U is, the higher the sensitivity of the device is. The

transfer function of the output voltage to the ground vibration acceleration is

$$H(s) = \frac{Us}{s^2 + 2D\omega_0 s + \omega_0^2}$$    (3)

**3 Reference corner cube inertial suspension device design**

The reference corner cube inertial suspension device (RCCISD) adopts the mechatronics design of the force balance principle, including the reference corner cube M, elastic suspension elements, transducers,

amplifiers, working coils and output circuits (Li Caihua et al., 2018).

The inertia force caused by the ground motion makes the reference corner cube M deviate from the equilibrium position. Then, the deviate drives the coil cutting magnetic field to output the voltage signal and produce an electromagnetic force acting on the M. The direction of the electromagnetic force is opposite to the inertia force felt by the M, whose magnitude is basically the same to the

electromagnetic force, so that the motion amplitude of the M is as small as possible. Because the inertia force felt by the M is proportional to the speed of the ground motion and the electromagnetic force acting on the M is proportional to the current passing through the electromagnetic coil, an electromechanical coupling system is formed. So, the electrical signal output proportional to the ground motion is obtained.

The composition and structure of the RCCISD is shown in figures 5. The mechanical display of the suspension reference corner cube is shown in figure 5, for example, where the elastic vibration system is composed of pendulum body (8 in figure 5), Reed (9 in figure 5), reference corner cube (10 in figure 5), etc.; the energy exchange system consists of a calibration coil (5 in figure 5), a magnetic cylinder (6 in figure 5), a working main coil (7 in figure 5), and other components (Teng Yuntian, 2001).



The elastic system adopts rotary compound pendulum structure, with cross-spring as rotation shaft. The leaf spring is used to be the main suspension, which is equipped with mechanical period adjusting screw and zero adjusting screw. The heavy hammer locking and the cross-spring locking are combined to the locking device. The physical figure is shown in figure 6, and the physical size is 20cm*16cm*20cm, which is about 1/3 of the super-long spring volume of the FG5 gravimeter.

**4 Experimental test**

**4.1 Data processing algorithm**

The vibration of the reference corner cube is homologous to the vibration information contained in the laser interference fringes. Since the vibration interference is coupled to the reference corner cube through the ground, the influence of the vibration can be evaluated through synchronous measurement 155 of the ground vibration.

**4.2 Shaking table experiment**

The RCCISD is tested on the shaking table(MA Jie-mei, et al..2014), and the test data are shown in Table 1.

The transfer function of the RCCISD is:

$$\left| H(s) \right|_{s=j\omega} = \left| \frac{A_0 s^2}{s^2 + 2D\omega_0 s + \omega_0^2} \right|_{s=j\omega} = \frac{-A_0 \omega^2}{\sqrt{\omega^4 - 2\omega_0^2 \omega^2 + 4D^2 \omega_0^2 \omega^2 + \omega_0^4}}$$
(4)

$\omega_0$ is the natural angular frequency of oscillation. $D$ is damping. $A_0$ is sensitivity.

Using the data in Table 1 for the transfer function fitting, the natural oscillation angular frequency is $\omega_0$ = 2.2942 $rad/s$, that is, natural oscillation frequency is $f_0$ = 0.3651 Hz. The sensitivity is $A_0$ = 387.7 $V/m/s$.The fitted amplitude-frequency characteristic curve is shown in 165 figure 8. In the picture, the red circle is the experimental data and the blue line is the fitting curve. The device shows high-pass characteristics.

**4.3 Transfer function calibration of the RCCISD**

Before the observation experiment, the RCCISD was sinusoidal calibrated with 17 calibration frequencies(LIN Zhan, et al.. 2015; ZHANG Xiao-peng, et al..2016; LV Yong-qing, et al.. 2020), as 170 shown in table 2 The calibration data is shown in figure 9.

Using the calibrated data, 1Hz was taken as the normalized frequency, and the normalized sensitivity of each frequency point was calculated in the frequency domain through FFT and ABS. The amplitude-frequency characteristics of the RCCISD could be obtained according to the calibrated data and sensitivity (XUE Bing. 2021), as shown in figure 7. The red is calibration test data and the blue is 175 the fitting curve. Through sinusoidal calibration, we can fit the more accurate transfer function of RCCISD in the experimental test, so as to obtain more accurate experimental data.





### 4.4 Joint experimental test of RCCISD and Age-110 laser interference absolute gravimeter

The joint observation experiment of Age-110 laser interference absolute gravimeter and RCCISD is carried out in the laboratory, as shown in figure 10. The gravimeter ran for one hour with the laser interference fringe data and vibration data from the RCCISD recorded(Wu Qiong, et al. 2012).

### 4.5 Vibration acceleration data processing

The algorithm in Section 4.1 was applied to obtain the vibration acceleration measured by the RCCISD, as shown in Figure 11. The periodic vibration signal of gravimeter during operation can be clearly seen. The coordinate in the picture is the data sampling sequence, and the sampling rate is 500sps, that is, the 500th data in the sequence is one second. It is obvious that the periodic vibration signal of the gravimeter is running, and the peak and peak value of the vibration acceleration is about $0.2m/s^2$ .

The vibration signal of free-falling corner cube during its one free fall is shown in figure 12. It can be seen that the process of a free-falling corner cube is static, free fall and deceleration(Wu Qiong. 2011;). The vibration caused by deceleration impact is maintained for about 0.5s.

### 4.6 Gravity acceleration solution

The interference fringes of Age-110 absolute gravimeter were collected. Each interference fringe lasts for about 0.6s, whose effective part is about 0.15s. The sequence of absolute gravitational acceleration values are calculated(T. Tsubokawa, et al.. 1999; KLOPPING, et al.. 1991; Qian, J., et al.. 2018; Svitlov S., et al.. 2010), as shown in figure 13. The corresponding sampling frequency is $500Hz$.

The gravity acceleration value sequence is solved by using an interference fringe to filter the vibration acceleration sequence in the free fall phase measured by the reference corner cube inertial suspension device, and the correlation is calculated, as shown in figure 14. it can be seen that the two waveforms are consistent in shape, but their amplitudes differ by an order of magnitude, and the vibration acceleration is one order of magnitude smaller than the g value sequence.

The instantaneous phase of the laser interference fringe signal is extracted based on the analytical signal processing algorithm, and the correlation analysis is carried out by combining the interference signal recorded by the single free fall of the falling body and the background vibration acceleration data directly placed on the ground by a wide band reference corner cube. It provides technical support for the establishment of the theoretical model of vibration interference suppression of absolute gravimeter. The method of vibration measurement and compensation requires high technical details and implementation technology. in the design and implementation, it is necessary to integrate the reference corner cube and vibration measurement system with other parts of the gravimeter. It is necessary to directly measure the vibration of the reference corner cube. Placing the reference corner cube on the pendulum of the elastic suspension of the seismograph can be directly measured and used for vibration compensation.

### 4.7 Quantitative analysis of vibration effect



Because the vibration interference is coupled to the reference corner cube through the ground, the vibration of the reference corner cube is the same as the vibration information contained in the laser interference fringes, so the result of the full waveform of the laser interference fringes is the same as the vibration mode of the reference corner cube, which is equivalent to measuring the vibration of the reference corner cube. The vibration and noise power spectral density (PSD) of the free fall process observed by the reference corner cube directly placed on the ground and the RCCISD is calculated, as shown in figure 15 (Peterson, J. R. 1993; XUE Bing. 2021; Bing Zhang, et al..2021). The blue is the PSD of gravity acceleration calculated by an interference fringe when the reference corner cube is placed directly on the ground. The pink is the PSD of gravity acceleration calculated by an interference fringe when the reference corner cube is placed on the RCCISD. The purple is the PSD of vibration noise recorded by the RCCISD during the gravimeter operation for one hour.

## 5 Discussion and Conclusions

It can be seen from figure 15 that the vibration and noise of the absolute gravimeter is higher than that of the earth's high noise model NHNM. When the reference corner cube is placed directly on the ground and observed by the inertia suspension device of the reference corner cube, the shape of the vibration and noise power spectrum in the process of free fall is similar. In the frequency band above $7Hz$, the vibration power of the reference corner cube jointly observed by the reference corner cube inertial suspension (RCCISD) is about $40dB$ lower than that of the reference corner cube directly placed on the ground. In other words, the RCCISD has a certain inhibition effect on high-frequency vibration interference, which is better than that of the reference corner cube directly placed on the ground. However, there is parasitic resonance around $150Hz$, which is recovered around 200Hz. Therefore, the effective observation band of the RCCISD needs to be extended to more than $200Hz$.

The FG5 absolute gravimeter uses a Super Spring vibration isolation system with a self-oscillation period of 60s to suppress vibration interference above $0.017Hz$ and achieve a measurement accuracy of $2\mu GAL/\sqrt{Hz}$(Micro-g LaCoste. 2015), corresponding to the ordinate $-154dB$. It can be seen from the PSD that in the $7.8Hz\sim50Hz$ frequency band, the lowest background noise power of the measuring point is about -150dB. It is necessary to solve the accuracy problem of vibration measurement in the development of RCCISD. The measurement accuracy of the gravimeter can be improved by 7 times after the average of 50 drops. If the measurement accuracy of FG5-X is $2\mu GAL/\sqrt{Hz}$, the accuracy of a single measurement should reach $10\mu GAL$, that is, $10^{-7}m/s^2$. According to the effective noise value of NHNM in the frequency band of $0.02Hz\sim1Hz$ is $1\times10^{-5}m/s^2$, it can be seen that the minimum noise suppression of background vibration should be $40dB$.

**Funding**: This research was funded by "Hebei Key Laboratory of Seismic Disaster Instrument and Monitoring Technology(Grant NO.FZ224201)" ,"National Key Research and Develop-ment Project",grant number 2022YFC2204301 and "the Special Fund of the Institute of Earthquake Forecasting, China Earthquake Administration",grant number CEAIEF2022030105.

**Data Availability Statement**: The observed data related to this study is uploaded in the open-access repository Zenodo for sharing purpose (https://doi.org/10.5281/zenodo.4785127).



**Conflicts of Interest**: The authors declare no conflict of interest.

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

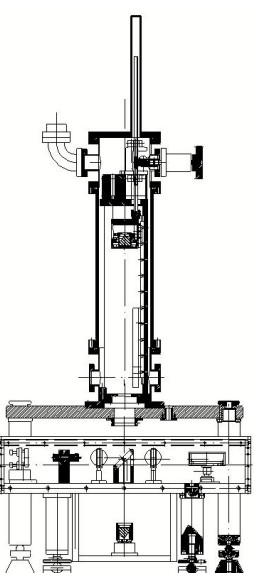

**Figure 1. Mechanical structure.**





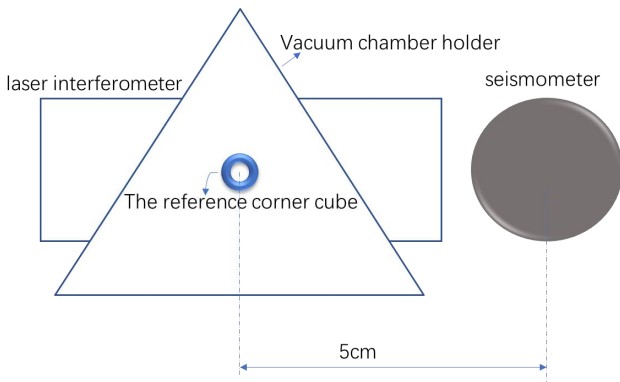

**Figure 2. Schematic diagram of synchronous observation position of seismograph.**

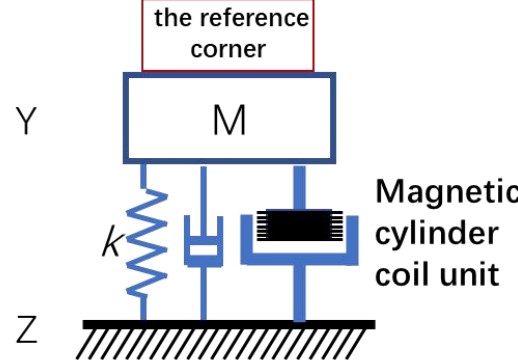

**Figure 3. Principle model.**

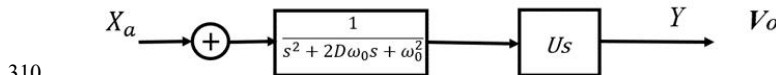


**Figure 4. The transfer function.**



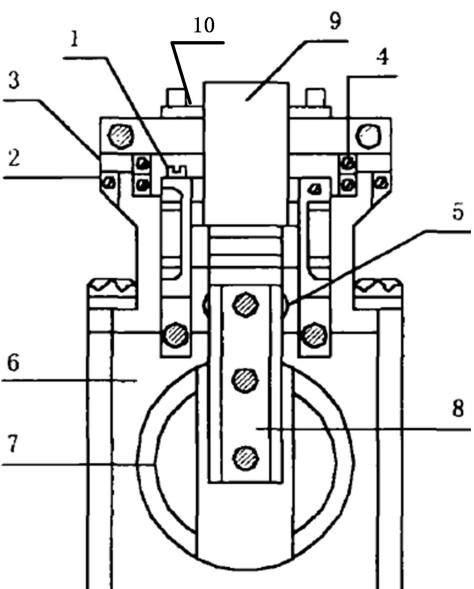

1 cycle adjusting screw 2 locking screw 3 locking screw 4 zeroing screw 5 calibration coil 6 magnetic cylinder 7 working main coil 8 pendulum 9 leaf spring 10 reference corner cube

**Figure 5. The composition of the RCCISD.**

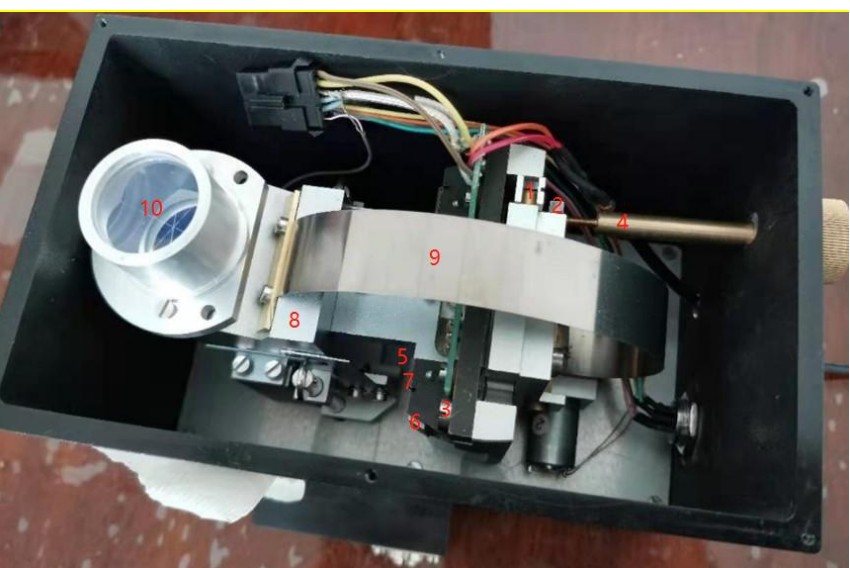

**Figure 6. Picture of the RCCISD.**





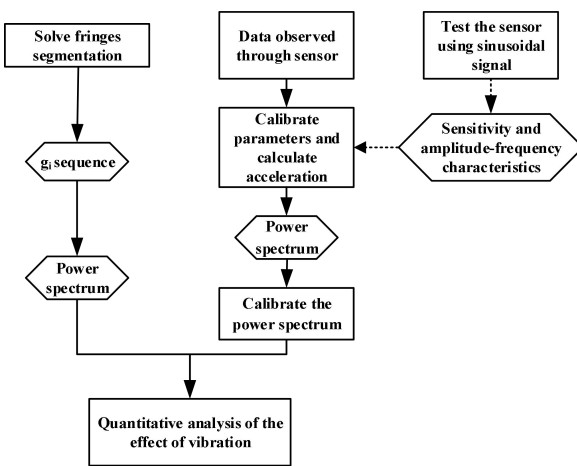

**Figure 7. Data processing flow.**

**Table1 Shaking table test results**

| Frequency (Hz) | Sensitivity (V/m/s) |
|---|---|
| 5 | 365.896 |
| 2 | 290.619 |
| 1 | 190.868 |
| 0.5 | 104.905 |
| 0.2 | 43.272 |
| 0.1 | 15.721 |



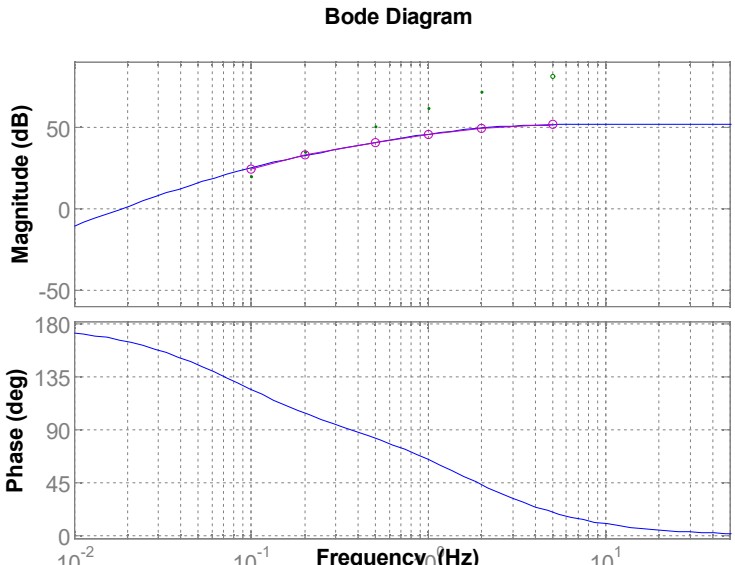

**Figure 8. The fitted amplitude-frequency characteristic curve**

**Table2 Calibration frequency**

| NO. | Cycle number(unit) | Frequency($Hz$) | Attenuation factor |
|---|---|---|---|
| 1 | 10 | 1/30 | 1000 |
| 2 | 100 | 0.05 | 1000 |
| 3 | 100 | 0.1 | 1000 |
| 4 | 100 | 0.2 | 1000 |
| 5 | 100 | 0.25 | 1000 |
| 6 | 100 | 1/3 | 1000 |
| 7 | 100 | 0.5 | 1000 |
| 8 | 200 | 1 | 1000 |
| 9 | 200 | 2 | 1000 |
| 10 | 200 | 3 | 1000 |
| 11 | 200 | 4 | 1 |
| 12 | 200 | 5 | 1 |
| 13 | 200 | 6 | 1 |
| 14 | 200 | 7 | 1 |





| 15 | 200 | 8 | 1 |
|----|-----|---|---|
| 16 | 200 | 9 | 1 |
| 17 | 200 | 10 | 1 |


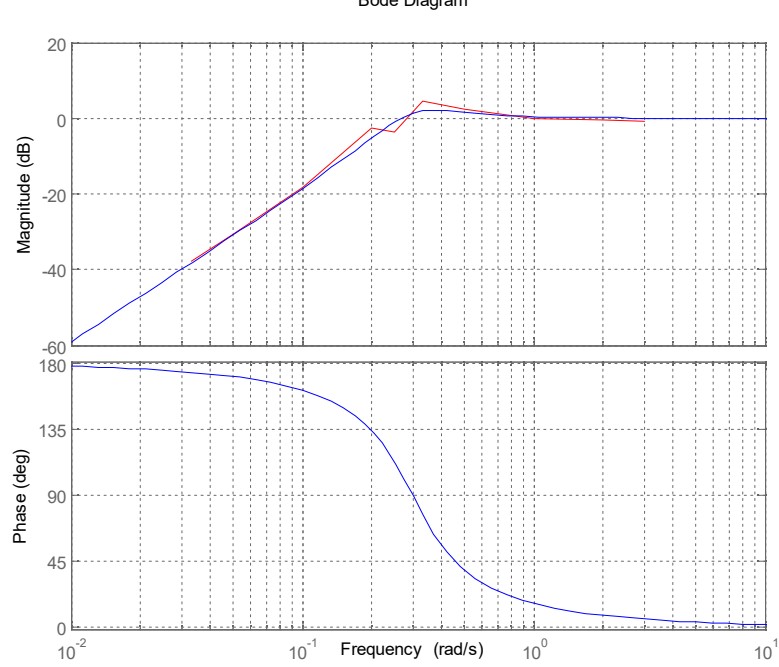

**Figure 9. Calibrated transfer function**

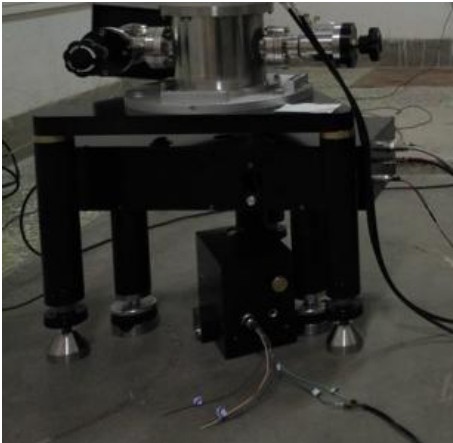

**Figure 10. Joint experimental test**



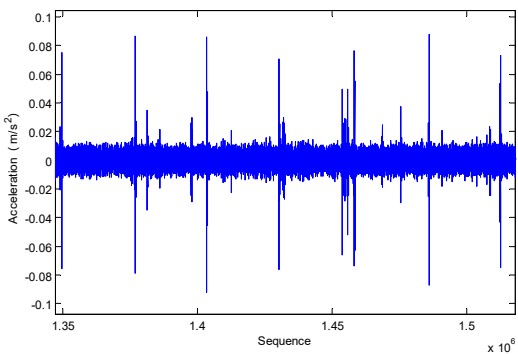

**Figure 11. Vibration acceleration data**

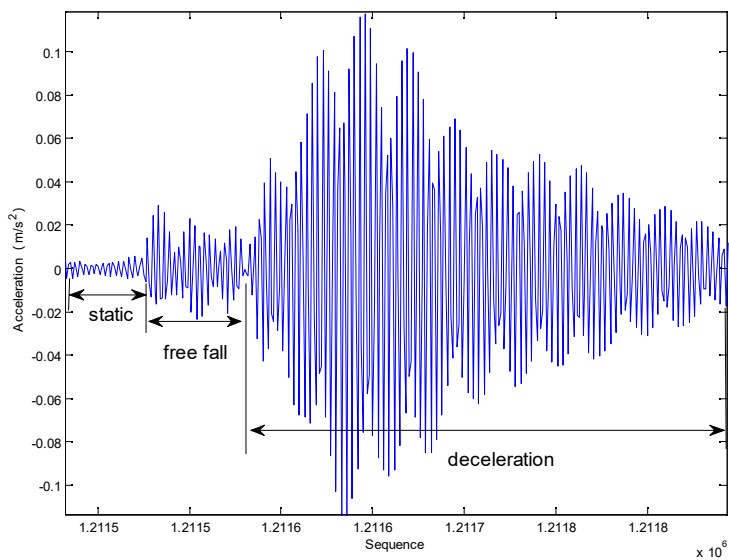

**Figure 12. Vibration data during a free fall**





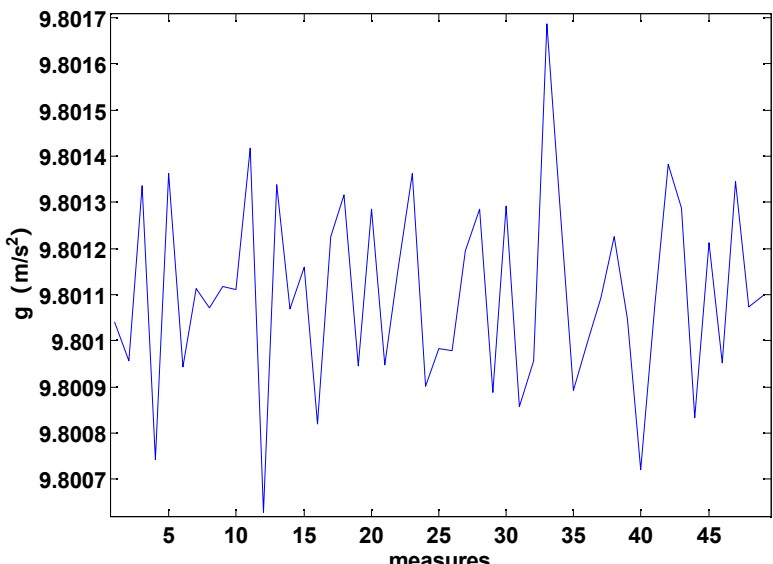


**Figure 13. The sequence of absolute gravitational acceleration**

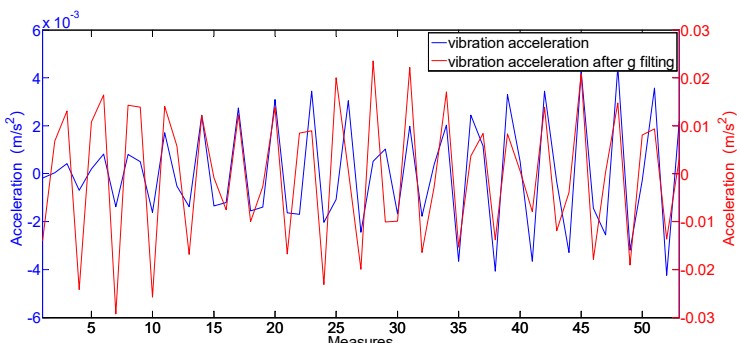

**Figure 14. Correlation calculation**

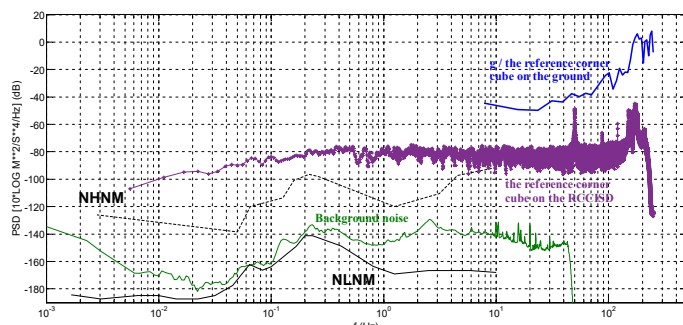

**Figure 15.The PSD of experimental data**