# Peer review of "The development of a reference corner cube inertial suspension device"

_Geoscientific Instrumentation, Methods and Data Systems, 2023_

## Referee Comment (RC1)

1 In this paper,the author introduces two transfer fucntions of the RCCISD. The first one is transfer fuction using the shake table test data (see section 4.2 ). The second one is transfer fuction using sinusoidal calibration data(see section 4.3 ). There are difference between these two transfer fuctions, especially from bandwidth 0.1Hz to 10Hz. Why use these two transfer fuctions in this paper? Does these two transfer functions have an impact on subsequent gravity acceleration measurements and experimental data analysis?

2 In section 4.4, during the experimental, where is the RCCISD? Be placed inside the vacuum chamber of the gravimeter? Or just be placed on the ground outside the vacuum chamber of the gravimeter? When the RCCISD was placed on ground outside the vacuum chamber of the gravimeter, Is the vibration acceleration data measured at different positions consistent? See following figure, near the A-pillar or B-pillar in the figure, the amplitude of vibration acceleration may be larger? In the figure, the vibration amplitude at the center position of the four pillar A, B, C, and D is smaller. But the small changes in the vibration acceleration signal measured by RCCISD may have a significant impact on the analysis of the gravity acceleration results of the gravimeter.

[Figure]

3 There are some unclear sentences in the article that need to be carefully revised.

seismometer does not coincide with the reference corner cube in space, resulting in spatial dislocation,
15      which cannot accurately reflect the vibration state of the reference corner cube. So, it is necessary to
        hang the reference corner cube on the elastic element to directly measure its vibration acceleration
        measurement. In this paper, an open-loop reference corner cube inertial suspension device(RCCISD)
        hanging the reference corner cube was developed based on the principle of seismometer, which is used

the reference corner cube directly placed on the ground. RCCISD can restrain the vibration interference
        to a certain extent, not only can it measure the reference corner cube vibration more accurately than the
        seismograph synchronous observation method for the vibration compensation of gravity measurement,
25      but also the volume is about 1 / 3 of the Super-Spring volume, which can greatly reduce the height of
        the gravimeter.

new open-loop reference corner cube inertial suspension device. The motion of the pendulum is measured by an electromagnetic transducer. A coil is installed on the pendulum and embedded in the gap magnetic field of the magnetic steel. The induced electromotive force is generated by the movement of the coil in the magnetic field, and the output of the voltage signal is used to reflect the

$$e_v = 2 \pi rNB \frac{dx}{dt} = U \frac{dx}{dt} \tag{2}$$

Where U is the voltage sensitivity, the larger U is, the higher the sensitivity of the device is. The transfer function of the output voltage to the ground vibration acceleration is

$$H(s) = \frac{Us}{} \tag{3}$$

---

## Author Comment (AC1)

**1 In this paper,the author introduces two transfer functions of the RCCISD. The first one is transfer function using the shake table test data (see section 4.2). The second one is transfer function using sinusoidal calibration data (see section 4.3 ). There are difference between these two transfer functions, especially from bandwidth 0.1Hz to 10Hz. Why use these two transfer functions in this paper? Do these two transfer functions have an impact on subsequent gravity acceleration measurements and experimental data analysis?**

**Answer 1:In section 4.2, The transfer fuction of the RCCISD using the shake table test data is used to roughly determine the natural frequency and frequency response curve of RCCISD, and to provide a reference for setting parameters for sinusoidal calibration.**
**In section 4.3, The main purpose of sinusoidal calibration test is to more accurately fit the transfer function of RCCISD in the joint experiment of gravimeter and RCCISD, so as to facilitate the more accurate removal of the frequency response of RCCISD in the subsequent gravity acceleration measurement data processing.**
**In response to your questions, I have added a description of the relevant content. As shown below**

The transfer function of the RCCISD is:

$$|H(s)|_{s=j\omega} = \left| \frac{A_0 s^2}{s^2 + 2D\omega_0 s + \omega_0^2} \right|_{s=j\omega} = \frac{-A_0\omega^2}{\sqrt{\omega^4 - 2\omega_0^2\omega^2 + 4D^2\omega_0^2\omega^2 + \omega_0^4}} \qquad (5)$$

$\omega_0$ is the natural angular frequency of oscillation. $D$ is damping. $A_0$ is sensitivity. Using the data in Table 1 for the transfer function fitting, the natural oscillation angular frequency is $\omega_0 = 2.2942 rad/s$ , that is, natural oscillation frequency is $f_0 = 0.3651 Hz$ . The sensitivity is $A_0 = 387.7V/m/s$ 。 The fitted amplitude-frequency characteristic curve is shown in figure 8. In the picture, the red circle is the experimental data and the blue line is the fitting curve. The device shows high-pass characteristics.

(Add new content:The transfer function of the RCCISD using the shake table test data is used to roughly determine the natural frequency and frequency response curve of RCCISD, and to provide a reference for setting parameters for sinusoidal calibration.)

Using the calibrated data, 1Hz was taken as the normalized frequency, and the normalized sensitivity of each frequency point was calculated in the frequency domain through FFT and ABS. The amplitude-frequency characteristics of the RCCISD could be obtained according to the calibrated data and sensitivity (XUE Bing. 2021), as shown in figure 7. The red is calibration test data and the blue is the fitting curve. Through sinusoidal calibration, we can fit the more accurate transfer function of RCCISD in the experimental test, so as to obtain more accurate experimental data. (Modifed to:The red is calibration test data and the blue is the fitting curve. Through sinusoidal calibration, we can fit the more accurate transfer function of RCCISD in the experimental test, so as to facilitate the more accurate removal of the frequency response of RCCISD in the subsequent gravity acceleration measurement data processing.)

**2 In section 4.4, during the experimental, where is the RCCISD? Be placed inside the vacuum chamber of the gravimeter? Or just be placed on the ground outside the vacuum chamber of the gravimeter? When the RCCISD was placed on ground outside the vacuum chamber of the gravimeter, Is the vibration acceleration data measured at different positions consistent? See following figure, near the A-pillar or B-pillar in the figure, the amplitude of vibration acceleration may be larger? In the figure, the vibration amplitude at the center position of the four pillar A, B, C, and D is smaller. But the small changes in the vibration acceleration signal measured by RCCISD may have a significant impact on the analysis of the gravity acceleration results of the gravimeter.**

**Answer 2:In section 4.4, during the experiment, the RCCISD is placed directly on the ground outside the vacuum of the gravimeter. Although the vibration acceleration data of the reference corner cube measured at different positions are different, the experimental scheme in this paper mainly measures the vibration acceleration data of the reference corner cube where the reference corner cube is located during the free fall of the free-falling corner cube. The data is the result of the comprehensive influence of the vibration of A, B, C, D and other positions of the cement pier in the figure, therefore, when the free-falling corner cube falls freely, the vibration of any position except the vibration of the reference corner cube is of no use to the algorithm in this paper.**

**3 There are some unclear sentences in the article that need to be carefully revised.**

**Answer 3:The unclear sentences in the article have benn carefully revised.**

**Abstract:** The seismometer synchronous observation and zero crossing methods are applied to laser interferometer absolute gravimeter to suppress the vibration interference. However, during the synchronous observation of the seismometer and the gravimeter, the observation point of the seismometer does not coincide with the reference corner cube in space, resulting in spatial dislocation, which cannot accurately reflect the vibration state of the reference corner cube. So, it is necessary to hang the reference corner cube on the elastic element to directly measure its vibration acceleration measurement. (1 Modifed to:However, during the synchronous observation of the seismometer and the gravimeter, the observation point of the seismometer does not coincide with the reference corner cube in space, resulting in spatial dislocation and impossibility to accurately reflect the vibration state of the reference corner cube. So it can be considered to accurately measure the vibration acceleration of the reference corner cube by inertial suspension.) In this paper, an open-loop reference corner cube inertial suspension device(RCCISD) hanging the reference corner cube was developed based on the principle of seismometer, which is used to measure the vibration acceleration of the reference corner cube of the laser interferometer absolute gravimeter. Experimental test results show that the power spectrum of gravitational acceleration calculated by an interference fringe observed jointly by the RCCISD is about 40dB lower than that of the reference corner cube directly placed on the ground. RCCISD can restrain the vibration interference to a certain extent, not only can it measure the reference corner cube vibration more accurately than the seismograph synchronous observation method for the vibration compensation of gravity measurement, but also the volume is about 1/3 of the Super-Spring volume, which can greatly reduce the height of the gravimeter. (2 Modifed to:The RCCISD can restrain the vibration interference to a certain extent. At the same time, it can not only measure the vibration of the reference corner cube more accurately than the seismograph synchronous observation method, but also the volume is about 2/3 smaller than the Super-Spring, so it can greatly reduce the height of the gravimeter.)

**2. Principle model**

In order to directly measure and analyze the vibration data of the reference corner cube, the reference corner cube can be elastically suspended on the pendulum of the open-loop seismometer to design a new open-loop reference corner cube inertial suspension device. The motion of the pendulum is measured by an electromagnetic transducer. A coil is installed on the pendulum and embedded in the gap magnetic field of the magnetic steel. (3 Modifed to:A coil is installed on the inertia magnetic bar to form the magnetic coil unit. The magnetic coil unit then is embedded into the gap of the magnetic cylinder to form the magnetic cylinder coil unit.) The induced electromotive force is generated by the movement of the coil in the magnetic field, and the output of the voltage signal is used to reflect the acceleration of the frame to the ground motion. The principal model is shown as figure 3.

Therefore, the induced electromotive force can be written as

$$e_v = 2\pi r N B \frac{dx}{dt} = U \frac{dx}{dt} \tag{2}$$

Where U is the voltage sensitivity, the larger U is, the higher the sensitivity of the device is. (4 Modifed to:Where U is the voltage sensitivity.The larger the U, the higher the sensitivity of the device.) The transfer function of the output voltage to the ground vibration acceleration is

$$H(s) = \frac{Us}{s^2 + 2D\omega_0 s + \omega_0^2} \tag{3}$$

---

## Author Comment (AC2)

**Table1** Shaking table test results

| Frequency(Hz) | Sensitivity(V/m/s) |
|---|---|
| 5 | 365.896 |
| 2 | 290.619 |
| 1 | 190.868 |
| 0.5 | 104.905 |
| 0.2 | 43.272 |
| 0.1 | 15.721 |

**Table2** Calibration frequency

| NO. | Cycle number(unit) | Frequency(Hz) | Attenuation factor |
|---|---|---|---|
| 1 | 10 | 1/30 | 1000 |
| 2 | 100 | 0.05 | 1000 |
| 3 | 100 | 0.1 | 1000 |
| 4 | 100 | 0.2 | 1000 |
| 5 | 100 | 0.25 | 1000 |
| 6 | 100 | 1/3 | 1000 |
| 7 | 100 | 0.5 | 1000 |
| 8 | 200 | 1 | 1000 |
| 9 | 200 | 2 | 1000 |
| 10 | 200 | 3 | 1000 |
| 11 | 200 | 4 | 1 |
| 12 | 200 | 5 | 1 |
| 13 | 200 | 6 | 1 |
| 14 | 200 | 7 | 1 |
| 15 | 200 | 8 | 1 |
| 16 | 200 | 9 | 1 |
| 17 | 200 | 10 | 1 |

[Figure]

**Figure 7.** Data processing flow.